# Empirics on the Expressiveness of Randomized Signature

**Enea Monzio Compagnoni**
Department of Mathematics
ETH Zürich, Switzerland
menea@ethz.ch

**Luca Biggio**
Department of Computer Science
ETH Zürich, Switzerland
luca.biggio@inf.ethz.ch

**Antonio Orvieto**
Department of Computer Science
ETH Zürich, Switzerland
antonio.orvieto@inf.ethz.ch

## Abstract

Time series analysis is a widespread task in Natural Sciences, Social Sciences and Engineering. A fundamental problem is finding an expressive yet efficient-to-compute representation of the input time series to use as a starting point to perform arbitrary downstream tasks. In this paper, we build upon recent work using *signature* of a path as a feature map, and investigate a computationally efficient technique to approximate these features based on linear *random* projections. We present several theoretical results to justify our approach, we analyze and showcase its empirical performance on the task of learning a mapping between the input controls of a Stochastic Differential Equation (SDE) and its corresponding solution. Our results show that the representational power of the proposed random features allows to efficiently learn the aforementioned mapping.

## 1 Introduction

Modeling time series is a common task in finance, physics, and engineering. Differential (or Difference) equations are the most widespread approach, as they allow to specify the controls of the dynamics (i.e. external fields, time, input signals, noise, etc.) as well as their precise influence.

Unfortunately, guessing the true algebraic expression of such equations from first principles considerations is often impossible. Given a dataset of several input-output pairs, one can approach the problem from a data-driven perspective and construct a surrogate of the underlying operator mapping controls (inputs) to output. However, the size of the available datasets is not often sufficiently large to support the requirements of modern over-parametrized deep learning algorithms [Brown et al., 2020, Kaplan et al., 2020]. Additionally, these techniques do not come with guarantees on their ability to extract provably robust features from the input paths, and their training can be highly computationally demanding.

In this work, we build on ideas from rough path theory and reservoir computing [Friz and Hairer, 2020, Schrauwen et al., 2007] to extract a highly expressive set of (random) features from the input controls, allowing efficient learning of the dynamics of complicated SDEs. The features we consider are obtained by integrating over a fixed time window a latent SDE whose dynamic is modulated by linear *random* vector fields and driven by the input controls. The output, i.e. the solution to the original SDE, is simply given by a trained linear projection of the random features.

35th Conference on Neural Information Processing Systems (NeurIPS 2021), Sydney, Australia.

This approach presents several advantages: a) Our random features provably approximate the *signature* of the control paths. In particular, we circumvent their potentially expensive calculation (see e.g. Lyons [2014], Kidger and Lyons [2020]) preserving their information content and geometrical properties (see also discussion in the next section); b) The features we extract are non-local, in the sense that they are the result of the evolution of an SDE over a fixed time window and therefore encode information about the whole considered time interval; c) Contrarily to many deep neural networks for time series analysis, our approach is not limited to uniformly sampled sequences [Fawaz et al., 2019].

This paper is structured as follows: In Section 2, we present some theoretical insights motivating our method, in Section 3, we empirically validate the performance of our method on various simulated datasets, in Section 4 with link the proposed approach with related works in the literature, and in Section 5, we discuss our conclusion and possible future work.

## 2 Background

### 2.1 Randomized Signature of a Path

Let $X : [0, T] \to \mathbb{R}^d$ be a continuous piecewise smooth $d-$dimensional path $X = \left(X^1, \cdots, X^d\right)$. We will refer to $X$ as the *control* and to its single components $X_i$ as *controls*. We denote by $\{e_1, \ldots, e_d\}$ the canonical basis of $\mathbb{R}^d$.

**Definition 1 (Signature)** *For any $t \in [0, T]$, the Signature of a continuous piecewise smooth path $X : [0, T] \to \mathbb{R}^d$ on $[0, t]$ is the countable collection $\mathbf{S}_{[0,t]}^X := \left(1, S_{[0,t]}^{X,1}, S_{[0,t]}^{X,2}, \ldots\right) \in \prod_{k=0}^{\infty} \left(\mathbb{R}^d\right)^{\otimes k}$ where, for each $k \geq 1$, the entries $S_{[0,t]}^{X,k}$ are the iterated integrals defined as*

$$S_{[0,t]}^{X,k} := \sum_{(i_1,\ldots,i_k)\in\{1,\ldots,d\}^k} \left(\int_{0\leq s_1\leq\cdots\leq s_k\leq t} dX_{s_1}^{i_1}\ldots dX_{s_k}^{i_k}\right) e_{i_1}\otimes\cdots\otimes e_{i_k}.$$

*We define the Truncated Signature of $X$ of order $M \geq 0$ as $\mathbf{S}_{[0,t]}^{X,M} := \left(1, S_{[0,t]}^{X,1}, \ldots, S_{[0,t]}^{X,M}\right) \in \prod_{k=0}^{M} \left(\mathbb{R}^d\right)^{\otimes k} =: \mathcal{T}^M\left(\mathbb{R}^d\right).$*

**Remark 1** *We highlight that $S_{[0,t]}^{X,k}$ lives in $\left(\mathbb{R}^d\right)^{\otimes k}$, which is the space of tensors of shape $(d, \ldots, d)$ given by $\mathbb{R}^d \otimes \cdots \otimes \mathbb{R}^d$ for $k$ times.*

**Theorem 1 (Signature is a Reservoir)** *Let $V_i : \mathbb{R}^m \to \mathbb{R}^m, i = 1, \ldots, d$ be vector fields regular enough such that $dY_t = \sum_{i=1}^{d} V^i\left(Y_t\right) dX_t^i, Y_0 = y \in \mathbb{R}^m$, admits a unique solution $Y_t : [0, T] \to \mathbb{R}^m$. Then, for any smooth test function $F : \mathbb{R}^m \to \mathbb{R}$ and for every $M \geq 0$ there is a time-homogeneous linear operator $L : \mathcal{T}^M\left(\mathbb{R}^d\right) \to \mathbb{R}$ which depends only on $(V_1, \ldots, V_d, F, M, y)$ such that $F\left(Y_t\right) = L\left(\mathbf{S}_{[0,t]}^{X,M}\right) + \mathcal{O}\left(t^{M+1}\right)$, and $t \in [0, T]$.*

This theorem suggests the first $M$ entries of the signature of $X$ are sufficient to linearly explain the solution of any differential equation driven by it. Unfortunately, calculating $\mathbf{S}_{[0,t]}^{X,M}$ requires the calculation of $\frac{d^{M+1}-1}{d-1}$ iterated integrals – which in total quickly becomes computationally expensive. Of course several techniques have been developed to circumvent this problem, in particular kernelization techniques, see, e.g., Kidger and Lyons [2020]. The next result provides a practical description of how it is possible to reduce the computational burden without losing too much explanatory power. The formal statements are provided in Appendix.

**Theorem 2 (Randomized Signature (Informal))** *For any $k \in \mathbb{N}$ big enough and appropriately chosen random matrices $A_1, \ldots, A_d$ in $\mathbb{R}^{k\times k}$ and random shifts $b_1, \ldots, b_d$ in $\mathbb{R}^{k\times 1}$, and any fixed sigmoid function $\sigma$, the solution of*

$$dZ_t^X = \sum_{i=1}^{d} \sigma\left(A_i Z_t^X + b_i\right) dX_t^i, \quad Z_0^X = (1, 0, \cdots, 0) \in \mathbb{R}^k, \quad t \in [0, T] \tag{1}$$

*– called the Randomized Signature of $X$ – has comparable approximation power as signature itself.*

To conclude, we highlight that the computational complexity of calculating $Z_t^X$ is $k^2 \times d$ for each time step.

## 3 Experiments

### 3.1 1-Dimensional Stochastic Double Well

Let us recall that the dynamics of the 1-Dimensional Stochastic Double Well process is given by

$$dY_t = \theta Y_t \left( \mu - Y_t^2 \right) dt + \sigma dW_t, \quad Y_0 = y_0 \in \mathbb{R}, \quad t \in [0, 1]$$

where $W_t$ is a 1-dimensional Brownian motion, and $(\mu, \theta, \sigma) \in \mathbb{R} \times \mathbb{R}^+ \times \mathbb{R}^+$. Let us fix $y_0 = 1$ and $(\mu = 2, \theta = 1, \sigma = 1)$, and the partition $\mathcal{D}$ of $[0, 1]$ to have $N = 101$ equally spaced time steps. We train a Ridge Regression with regularization parameter $\lambda = 0.001$ to map instances of Randomized Signature of the controls $Z^{[t, W_t]}$ into the respective solution $Y_t$. We repeat the experiment on different values of the number $N_{\text{Train}}$ of train samples and dimension $k$ of the $Z^{[t, W_t]}$. On the left of Figure 1, we plot an example of the trajectory of $Z^{[t, W_t]}$ while, on its right, we plot the comparison of the true and the generated time series on an out of sample case. The following table shows the performance in terms of $L^2$ relative error on 10000 test samples:

| | $N_{\text{Train}} = 1$ | $N_{\text{Train}} = 10$ | $N_{\text{Train}} = 100$ | $N_{\text{Train}} = 1000$ | $N_{\text{Train}} = 10000$ |
|---|---|---|---|---|---|
| $k = 111$ | **0.197858** | **0.029245** | **0.015954** | 0.007171 | 0.005397 |
| $k = 222$ | 0.322285 | 0.039826 | 0.023862 | **0.0066585** | 0.0052776 |
| $k = 708$ | 0.363072 | 0.321875 | 0.069075 | 0.010936 | **0.0050793** |

Table 1: Double Well: Relative $L^2$ Error

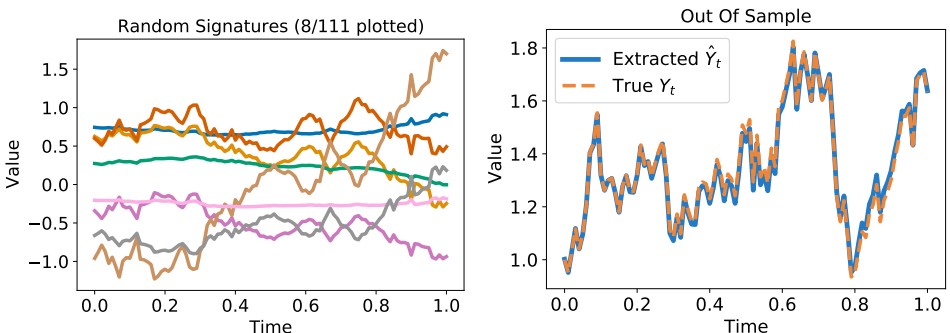

Figure 1: Double Well: Random Signatures (left) - Test Sample (right)

### 3.2 4-Dimensional Ornstein–Uhlenbeck process

Let us recall that the dynamics of the 4-Dimensional Ornstein–Uhlenbeck process is given by

$$d\mathbf{Y}_t = (\boldsymbol{\mu} - \boldsymbol{\Theta} \mathbf{Y}_t) dt + \boldsymbol{\Sigma} d\mathbf{W}_t, \quad \mathbf{Y}_0 = \mathbf{y}_0 \in \mathbb{R}^4, \quad t \in [0, 1]$$

where $\mathbf{W}_t$ is a 4-dimensional Brownian motion, and $(\boldsymbol{\mu}, \boldsymbol{\Theta}, \boldsymbol{\Sigma}) \in \mathbb{R}^4 \times \mathbb{R}^{4 \times 4} \times \mathbb{R}^{4 \times 4}$. Let us fix $\mathbf{y}_0 = \mathbf{1}$, $\boldsymbol{\mu} = \mathbf{1}$, $\boldsymbol{\Sigma} = \mathbb{1}_4$, $\boldsymbol{\Theta}_{i.j} = i/j$, the partition $\mathcal{D}$ of $[0, 1]$ to have $N = 101$ equally spaced time steps, and $k = 708$. Finally, we train a Ridge Regression with $\lambda = 0.001$ on $N_{\text{Train}}$ train sample and Figure 2 shows the comparison of Out of Sample generated and true trajectories while Table 3 in Appendix reports the performance in terms of $L^2$ relative error on 10000 test samples.

### 3.3 1-Dimensional Stochastic Double Well - Irregularly Sampled Time Grid

In this experiment, we fix $y_0 = 1$ and $(\mu = 2, \theta = 1, \sigma = 1)$. On the other hand, for each train and test sample, the partition $\mathcal{D}$ of $[0, 1]$ is made of $N$ randomly drawn times. More precisely, $\mathcal{D} = \{0, t_1, \cdots, t_{N-1}, 1\}$ such that $t_k = 1/(1 - \exp(-s_k))$ and $\{s_1, \cdots, s_{N-1}\}$ are $N - 2$ independent realizations of a uniform distribution $\mathcal{U}[0, 1]$ sorted increasingly. As a result, the probability that two

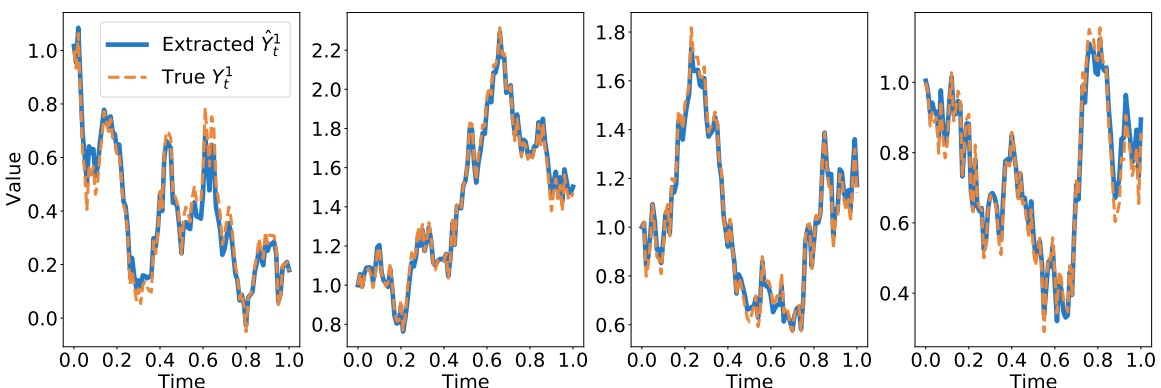

Figure 2: 4-Dimensional Ornstein–Uhlenbeck process: Test Sample

samples share the same $\mathcal{D}$ is null. We train a Ridge Regression with $\lambda = 0.001$ on $N_{\text{Train}} = 10000$ train samples and Figure 3 in Appendix shows the comparison of an Out of Sample generated and true trajectory. Finally, Table 2 shows the Relative $L^2$ Error on 10000 test samples as we vary the number of time steps $N$ and $k$ and compare it to the respective experiment in case the time grid is regularly spaced.

| | $(N, k) = (10, 56)$ | $(N, k) = (100, 222)$ | $(N, k) = (1000, 332)$ |
|---|---|---|---|
| Irregular | 0.073190 | 0.016717 | 0.007677 |
| Regular | 0.028876 | 0.005128 | 0.002763 |

Table 2: Irregularly Sampled Double Well: Relative $L^2$ Error

## 4   Related Works

**Random Features and Reservoir Computing.** The idea of extracting features based on random operations is not new and has seen a number of successful applications over the past years. On particular note for our method, the seminal work of Rahimi and Recht [2008] proposes to accelerate kernel machines by learning random features whose inner product matches that of a target shift-invariant kernel. The trade-off between generalization and computational efficiency of learning with random features has then been rigorously studied by Rudi and Rosasco [2017]. A conceptually very similar rationale is introduced by a parallel series of works exploring the topic of Reservoir Computing [Schrauwen et al., 2007]. Similarly to our work, Echo State Networks [Jaeger, 2003] evolve the input state by a series of fixed random projection (the reservoir) and generate the output by applying a trainable linear projection over the hidden states. However, we make the additional step of linking the random features to the signature of the input path and, in our case, the evolution of the features is dictated by the differential equation shown in equation 1.

**Controlled Differential Equations.** Our work is also related with a series of recent papers investigating the problems of how to process irregular time series and to condition a model on incoming information through the lens of controlled differential equations [Kidger et al., 2020, Morrill et al., 2021]. Differently from them, our method is way more efficient since the only parameters we need to train are those of the final (often) linear readout layer.

**Rough Path Theory.** Rough path theory is about describing how possibly highly oscillatory (rough) control path interact with nonlinear systems [Lyons, 2014]. The concept of signature is introduced in this context to provide a powerful description of the input path upon which the theory is built. Recent years have seen a resurgence of these ideas, which have been revisited from a machine learning perspective [Bonnier et al., 2019, Kidger and Lyons, 2020]. Our analysis is strongly influenced by the work of Cuchiero et al. [2021b], who firstly establishes a connection between reservoir computing and signature of a paths (in a discrete setting).

# 5    Conclusions

In this work, we empirically analyze an approach based on random features to find a powerful representation of an input path. Our randomly extracted features approximate the signature of the given path and thus find theoretical support in the theory of rough path. We empirically assess the effectiveness of our method by inspecting its performance on several tasks based on learning a map between possibly multivariate and irregularly sampled input controls into the solution of the corresponding SDEs.

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

# A   Appendix

## A.1   Basic Definitions

First of all, we define the concept of admissible tensor norms, which we assume to have in this work.

**Definition 2 (Admissible Tensor Norms)** *Let $E := \mathbb{R}^d$ and $\otimes$ be a tensor product such that the tensor powers of $E$, $\left(E^{\otimes k} : k \geq 1\right)$, are equipped with a family $(\|\cdot\|_{E^{\otimes k}} : k \geq 1)$ of norms satisfying:*

1. *For $j, k \in \mathbb{N}$ and for all $h \in E^{\otimes j}$ and $l \in E^{\otimes k}$*

$$\|h \otimes l\|_{E^{\otimes(j+k)}} \leq \|h\|_{E^{\otimes j}} \|l\|_{E^{\otimes k}};$$

2. *For any permutation $\sigma$ of $\{1, \ldots, k\}$,*

$$\|l_1 \otimes \ldots \otimes l_k\|_{E^{\otimes k}} = \left\|l_{\sigma(1)} \otimes \ldots \otimes l_{\sigma(k)}\right\|_{E^{\otimes k}};$$

3. *For any bounded linear functionals $f$ on $E^{\otimes j}$ and $g$ on $E^{\otimes k}$, there exists a unique bounded linear functional, denoted as $f \otimes g$, on $E^{\otimes(j+k)}$ such that for all $h \in V^{\otimes j}$ and $l \in E^{\otimes k}$*

$$f \otimes g(h \otimes l) = f(h)g(l).$$

*A family of tensor norms satisfying these conditions is called a family of admissible tensor norms.*

Then we define the space in which the signature of a path lies in, that is the following Tensor Algebra.

**Definition 3 (Tensor Algebra $\mathcal{T}\left(\mathbb{R}^d\right)$)** *We define the tensor algebra on $\mathbb{R}^d$ as*

$$\mathcal{T}\left(\mathbb{R}^d\right) := \prod_{k=0}^{\infty} \left(\mathbb{R}^d\right)^{\otimes k}$$

*as well as its truncated version of order $M \geq 0$ as*

$$\mathcal{T}^M\left(\mathbb{R}^d\right) := \prod_{k=0}^{M} \left(\mathbb{R}^d\right)^{\otimes k},$$

*where $\left(\mathbb{R}^d\right)^{\otimes k}$ is the space of tensors of shape $(d, \ldots, d)$ given by $\mathbb{R}^d \otimes \cdots \otimes \mathbb{R}^d$ for k times.*

**Definition 4 (Concatenation Operation)** *We define the Concatenation Operation $*$ such that for any given couple of continuous piecewise smooth paths $X : [0, s] \to \mathbb{R}^d$ and $Y : [s, t] \to \mathbb{R}^d$, their image through $*$ is the continuous piecewise smooth path $X * Y : [0, t] \to \mathbb{R}^d$ defined by*

$$X * Y_u := \begin{cases} X_u & \text{if } u \in [0, s] \\ X_s + Y_u - Y_s & \text{if } u \in [s, t] \end{cases}$$

**Definition 5 (Inverse Operation)** *We define the Inverse Operation $\leftarrow$ such that for any continuous piecewise smooth path $X : [0, T] \to \mathbb{R}^d$, its image through $\leftarrow$ is the continuous piecewise smooth path $\overleftarrow{X}_t := X_{T-t}$, for each $t \in [0, T]$.*

**Definition 6 (Topological Space)** *A topological space is an ordered pair $(X, \tau)$, where $X$ is a set and $\tau$ is a collection of subsets of $X$, satisfying the following axioms:*

1. *The empty set and $X$ itself belong to $\tau$;*

2. *Any arbitrary (finite or infinite) union of members of $\tau$ still belongs to $\tau$;*

3. *The intersection of any finite number of members of $\tau$ still belongs to $\tau$.*

*The elements of $\tau$ are called open sets and the collection $\tau$ is called a topology on $X$.*

**Definition 7 (Arcwise Connected Topological Space)** *A topological space $(X, \tau)$ is said to be arcwise connected if any two distinct points $x, y \in X$ can be joined by an arc, that is a continuous map $\alpha : [0, 1] \to X$ such that $\alpha(0) = x$ and $\alpha(1) = y$.*
*We say that $X$ is uniquely arcwise connected if for any two distinct points $x, y \in X$, there exists a unique path in $X$ that joins them.*

**Definition 8 ($\mathbb{R}$-tree)** *An $\mathbb{R}$-tree is a uniquely arcwise connected metric space, in which the arc between two points is isometric to an interval.*

**Definition 9 (Tree-Like)** *A continuous piecewise smooth path $x : [0, T] \to \mathbb{R}^d$ is tree-like if there exists an $\mathbb{R}$-tree $\tau$, a continuous map $\phi : [0, T] \to \tau$ and a map $\psi : \tau \to V$ such that $\phi(0) = \phi(T)$ and $x = \psi \circ \phi$.*

As T. Levy showed in Section 5.7 of [], a continuous piecewise smooth path is tree-like if and only if it is contractible to the constant path within its own image.

## A.2 Supporting Results

**Theorem 3** *Given a continuous piecewise smooth path $X : [0, T] \to \mathbb{R}^d$, its Signature is the banal signature $\mathbf{1} := (1, 0, 0, \cdots)$ if and only if $X$ is tree-like. In particular $S(X * \overleftarrow{X}) = 1$.*

**Lemma 4 (Johnson-Lindenstrauss Lemma)** *Given an $M$-dimensional Hilbert space $(H, \langle \cdot, \cdot \rangle_H)$ and $Q$ a $N$-point subset of $H$, for any $0 < \epsilon < 1$, for each $k \in \mathbb{N}$ satisfying the so called Johnson-Lindenstrauss constraint*

$$k \geq \frac{24 \log N}{3\epsilon^2 - 2\epsilon^3},$$

*there exists a linear map $f^{\{Q, M, k\}} : H \to \mathbb{R}^k$ that embeds $Q$ into $\mathbb{R}^k$ in an almost isometric manner. More specifically, we have that*

$$(1 - \epsilon) \|\mathbf{a}_1 - \mathbf{a}_2\|_H^2 \leq \left\| f^{\{Q, M, k\}}(\mathbf{a}_1) - f^{\{Q, M, k\}}(\mathbf{a}_2) \right\|^2 \leq (1 + \epsilon) \|\mathbf{a}_1 - \mathbf{a}_2\|_H^2$$

*for each $\mathbf{a}_1, \mathbf{a}_2 \in Q$.*

**Remark 2** *When there is no need to specify the dependence of $f^{\{Q, M, k\}}$ from $Q$, $M$, or $k$, we will omit them up to referring to is simply as $f$. Finally, we define $f^* : \mathbb{R}^k \to H$ to be the adjoint map of $f$ with respect to a fixed inner product $\langle \cdot, \cdot \rangle$ in $\mathbb{R}^k$.*

To apply such Lemma in our context, we select $M \geq 0$, equip $T^M(\mathbb{R}^d)$ with an inner product such that

$$\langle \mathbf{e}_{i_1} \otimes \cdots \otimes \mathbf{e}_{i_M}, \mathbf{e}_{j_1} \otimes \cdots \otimes \mathbf{e}_{j_M} \rangle := \delta_{i_1 j_1} \cdots \delta_{i_M j_M},$$

where $\{\mathbf{e}_{i_1} \otimes \cdots \otimes \mathbf{e}_{i_M}\}_{i_1, \ldots, i_M \in \{1, \ldots, d\}}$ is the canonical basis of $T^M(\mathbb{R}^d)$. Therefore, we have that $\left( T^M(\mathbb{R}^d), \langle \cdot, \cdot \rangle \right)$ is an Hilbert space.

**Theorem 5 (Existence and Uniqueness of the Signature)** *The following controlled differential equation*

$$d\mathbf{S}_t^X = \sum_{i=1}^d \mathbf{S}_t^X \otimes e_i dX_t^i, \quad \mathbf{S}_0^X = \mathbf{1}$$

*has a unique solution $\mathbf{S}_t^X$ which, at each $t \in [0, T]$ is the signature $\mathbf{S}_{[0,t]}^X$ of $X$ on $[0, t]$.*

**Notation 1** *In the light of the previous result, if there is no ambiguity about $[0, t]$, we will often refer to $\mathbf{S}_{[0,t]}^X$ as $\mathbf{S}^X$ and as $\mathbf{S}_t^X$ to stress its path-like nature. Analogously, we will use $\mathbf{S}^{X,M}$ and $\mathbf{S}_t^{X,M}$ in place of $\mathbf{S}_{[0,t]}^{X,M}$.*

Now, we provide results to show the relevance of these features. First of all, the following theorem ensures that $\mathbf{S}^X$ encodes the essence of $X$ and characterizes it completely.

According to Theorem 3, the concatenation $X * \overleftarrow{X}$ of $X$ with its inverse[1] $\overleftarrow{X}$ has the same signature as the constant path, but cannot be reparametrised to be constant. Similarly, if $X, Y, Z$ are non-constant paths, then $\mathbf{S}^{X*Y*\overleftarrow{Y}*Z*\overleftarrow{Z}*\overleftarrow{X}} = \mathbf{1}$, but $X * Y * \overleftarrow{Y} * Z * \overleftarrow{Z} * \overleftarrow{X}$ is not a path of the form $\gamma * \overleftarrow{\gamma}$ for any path $\gamma$. While the formal definition of tree-like path is given in Definition 10, Figure 1 guides our intuition as we notice that these paths look like trees and can be reduced to a constant path by removing possibly infinitesimal pieces of the form $\gamma * \overleftarrow{\gamma}$.

**Theorem 6 (Characterizing Nature of the Signature)** *Given a couple of continuous piecewise smooth paths $X$ and $\hat{X}$, then $\mathbf{S}^X = \mathbf{S}^{\hat{X}}$ if and only if $X * \overleftarrow{\hat{X}}$ is tree-like.*

This result is actually much stronger as it implies that the solution of any differential equation controlled by $X$ is fully determined by the vector fields and $\mathbf{S}^X$. In particular, Theorem 1 shows that the solution of any differential equation controlled by $X$ is essentially linear in $\mathbf{S}^X$.

Adapting Theorem III.7 in [Cuchiero et al., 2021b] it can be be shown that (asymptotically) the JL projected vector fields stem from random matrices:

**Theorem 7 ($Z^X$ is a random dynamical system)** *For $k \longmapsto \infty$, for each $i \in \{1, \cdots, d\}$ the linear vector fields $f^{\{k\}}\left(f^{\{k\}*}(\cdot)e_i\right) : \mathbb{R}^k \to \mathbb{R}^k$ are square matrices with asymptotically normally distributed, independent entries.*

Adapting Theorem III.8. [Cuchiero et al., 2021b] as done in [Cuchiero et al., 2021a] one obtains:

**Theorem 8 (Randomized Signature)** *For any fixed integer $M \geq 0$, any fixed partition $\mathcal{D} = \{t_1, \cdots, t_N\}$ of $[0, T]$ such that $0 \leq t_1 < \cdots < t_N \leq T$, let us consider the Truncated Signature of $X$ at such times, that is $Q := \left\{S_{t_1}^{X,M}, \cdots, S_{t_N}^{X,M}\right\}$ such that its elements all lie in $\left(T^M\left(\mathbb{R}^d\right), \langle\cdot, \cdot\rangle\right)$. Let us now select $0 < \epsilon < 1$, $k \in \mathbb{N}$ satisfying the associated Johnson-Lindenstrauss constraint, let $f$ be the implied Johnson-Lindenstrauss map and $f^*$ its adjoint map. Then, the solution of the controlled differential equation in $\mathbb{R}^k$*

$$dZ_t^X = \sum_{i=1}^d f\left(f^*\left(Z_t^X\right)e_i\right)dX_t^i, \quad Z_0^X = f(\mathbf{1}),$$

*on $\mathcal{D}$, that is $\left\{Z_{t_1}^X, \cdots, Z_{t_N}^X\right\}$, are called the Randomized Signature of $X$ on $\mathcal{D}$. Each $Z_{t_k}^X$ is the projection of $S_{t_k}^{X,M}$ from $\left(T^M\left(\mathbb{R}^d\right), \langle\cdot, \cdot\rangle\right)$ onto $\mathbb{R}^k$ via $f$ and thus preserves its geometric properties and its approximation power.*

**Definition 10 (Localized Randomized Signature)** *For any random matrices $A_1, \ldots, A_d$ in $\mathbb{R}^{k \times k}$ and shifts $b_1, \ldots, b_d$ in $\mathbb{R}^{k \times 1}$ such that maximal non-integrability holds on a starting point $z \in \mathbb{R}^k$, any fixed sigmoid function $\sigma$, and $d$-dimensional control $X$, the solution of*

$$dZ_t^X = \sum_{i=1}^d \sigma\left(A_i Z_t^X + b_i\right)dX_t^i, \quad Z_0^X = z, \quad t \in [0, T] \tag{2}$$

*is called the localized randomized signature of $X$.*

---

[1]$\overleftarrow{X}_t := X_{T-t}$

## A.3 Experiments: Supporting material

| $k = 708$ | $N_{\text{Train}} = 1$ | $N_{\text{Train}} = 10$ | $N_{\text{Train}} = 100$ | $N_{\text{Train}} = 1000$ | $N_{\text{Train}} = 10000$ |
|---|---|---|---|---|---|
| $Y^1$ | 0.375016 | 0.076250 | 0.011959 | 0.002199 | 0.001648 |
| $Y^2$ | 0.385306 | 0.096566 | 0.014064 | 0.002837 | 0.002002 |
| $Y^3$ | 0.353375 | 0.117444 | 0.016048 | 0.003129 | 0.001995 |
| $Y^4$ | 0.468581 | 0.091604 | 0.015304 | 0.002707 | 0.001814 |

Table 3: 4-Dimensional Ornstein–Uhlenbeck process: Relative $L^2$ Error

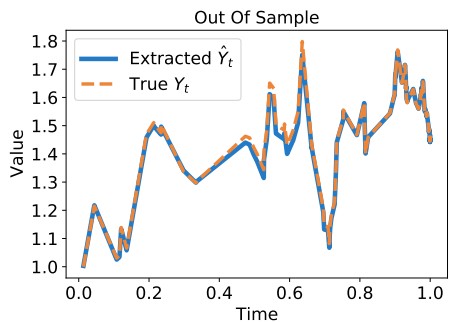

Figure 3: Irregularly Sampled Double Well: Test Sample

