# OpenReview forum: "Empirics on the expressiveness of Randomized Signature"
_NeurIPS.cc/2021/Workshop/DLDE — DLDE Workshop -- NeurIPS 2021 Poster_

### Official Review · Reviewer_TYFe · 2021-10-01
**Review for Empirics on the expressiveness of Randomized Signature**

**Confidence:** 2

**Review:**

In this paper the authors empirically analyze the use of an approach based on random features to find the representation of an input path. They use randomly extracted features to approximate the signature of the given path. The paper empirically tests the efficacy of the method by testing the performance on several tasks.

**Score:**

3: Good paper

---

### Official Review · Reviewer_scrT · 2021-10-05

**Confidence:** 2

**Review:**

# Summary
In this paper, the authors present an approach to approximate signatures using random projections.

The authors show empirically on simulated datasets that using sufficiently many of these random projections is enough for learning the dynamics of SDEs.

The method is the only approach presented in the paper and there is some work needed to sell the idea. Comparisons with other methods are needed to put this method into a proper context. The authors claim that the computational complexity of calculating $Z_t^X$ is $k^2\times d$ for each timestep $t$. How does that compare to other methods? A table to compare it with prior work would be very helpful.

With this criticism, I'm mainly trying to answer the question: why should I use this approach? What is the advantage compared to alternative approaches that can learn the dynamics of complicated SDEs? Does this method achieve a lower error or is it more efficient in some way?

You mention the work of Kidget et al. and Morrill et al. in section 4, can you compare their methods empirically with yours? Both on time complexity and generalization performance?

# Problems

Here are some minor comments.

In line 30: Your random features approximate the signatures of the control path but do you need less memory to work with them?

In line 40: With link? -> we link ... to related work...

In line 44 and 45: You use a superscript when you define $X$ and a subscript when you refer to $X_i$. Please fix this subscript one.

In line 62: ... in the Appendix

In line 66: Can you make the statement more precise? What does "comparable" mean here?

In Table 1: Why do you not test lower values of $k$?

**Score:**

2: Borderline paper

---

### Official Review · Reviewer_C83g · 2021-10-11
**Review: Good work, additional clarification welcomed**

**Confidence:** 2

**Review:**

**Summary:**

The paper is interested in approximating the solution of SDEs from continuous control inputs. The signature is a transformation (expansion) of these controls that allows to construct the solution to any SDE driven by these control via a time homogenous linear operator. The signature, even when truncated, is prohibitively hard to compute even in simple settings. The authors show that a relatively simple differential equation - whose solution is called the randomized signature - driven by the control inputs can achieve the same 'capacity' as the signature, in the sense of the ability to linearly reconstruct the solution of the SDE from it. The method is tested on a series of SDEs whose controls are sampled from Brownian processes. Overall the reported results are very convincing.

**Main review:**

I cannot comment in much depth on the mathematical contribution. However, I have clarification questions related to the implementation

1. The readout of the signature `Z` at time t  takes all entries of Z till `t : Z[0,t]`.  Are your readout weight thus the same size of the grid? Is this how you implemented it or do you use a sliding window (crop the past?)

2. Same question for the irregular grid: in this setting doing this regression does not really make sense: the readout should be grid-dependent.

If you could add some discussion on this point, that would help.

**Score:**

3: Good paper

---

### Official Review · Reviewer_oMjB · 2021-10-12

**Confidence:** 2

**Review:**

The authors propose an approach to extract a set of random features from the input controls of an SDE and train a linear projection on these random features as a solution to the original SDE. The features are integrals of a latent SDE over a fixed time window, where the randomness of the features is due to the SDE dynamics modulated by linear random vector fields.

It’ll be interesting to see how performance scales with the choice of $M$. Additional details should be included on the choice of $M$ and $T$ and their effects.

The proposed approach and experimental results should also be compared against, for example [Kidger et al., 2020, Morrill et al., 113 2021]. The efficiency claims should be quantified.


**Score:**

3: Good paper

---

### Decision · Program_Chairs · 2021-10-16

**Decision:**

Accept (Poster)

**Comment:**

Reviewers expressed mixed opinions about this paper. Comparisons to Kidger et al. 2020 and Morrill et al. 2021 were both mentioned as reasonable missing comparisons. I would additionally note that empirical comparisons to the truncated (non-random) signature would be beneficial. The number of terms in the signature scale exponentially with the dimension of the path, but the highest dimension considered here is 4, for which it is computationally tractable to compute the truncated signature up to some moderate depth.

I would additionally echo the comments of reviewer scrT, who ask "why should I use this approach?" For example it seems like all examples involve fitting an SDE whose entire trajectory is observed. This is unlike Kidger et al. 2020, who are able to learn "static" functions of a time series. (e.g. a classification result, not an evolving trajectory)

Despite these concerns I am inclined to accept the paper.

In addition to the comments made by the reviewers, I would make a number of my own:

- I believe there is a general lack of clarity in the presentation. For example, in Section 1: CDEs and signatures are usually to used model *functions of time series*, whilst SDEs are usually used to model *the time series itself*, but the introduction largely muddles these two concepts together. This lack of care extends to the rest of the paper, which I believe is relatively difficult to read unless one is already familiar with much of the material. I would strongly encourage the authors to approach their work from the point of view of the non-expert -- who may not be familiar with CDEs or signatures -- and seek to re-frame anything which may cause confusion. In my opinion this is actually the single greatest weakness of the paper.
- The definition of a controlled differential equation (first appearing in Theorem 1) is never introduced.
- Theorem 1 does not define the nature of (in particular the regularity of) the control $X$. No reference is given for this result, either.
- Section 3.1: The notation $Z^{[t, W_t]}$ may at first glance be mistaken for a typo of $Z^{W_t}_t$. That time must be treated as part of the control is one of the subtler aspects of CDEs/signatures, and deserves special mention.
- References: several published works are referred to only by their arXiv number.